# The Multi-Omics Analysis Revealed Microbiological Regulation of Rabbit Colon with Diarrhea Fed an Antibiotic-Free Diet

**DOI:** 10.3390/ani12121497

**Published:** 2022-06-08

**Authors:** Yang Chen, Jie Wang, Huimei Fan, Siqi Xia, Kaisen Zhao, Guanhe Chen, Yuchao Li

**Affiliations:** 1College of Animal Science and Technology, Sichuan Agricultural University, Chengdu 611130, China; chenyi154121@163.com (Y.C.); fanhuimei1998@163.com (H.F.); xiasiqi2020@163.com (S.X.); zhaofl0303@163.com (K.Z.); cgh_0132@163.com (G.C.); lichao_1116@163.com (Y.L.); 2Farm Animal Genetic Resources Exploration and Innovation Key Laboratory of Sichuan Province, Sichuan Agricultural University, Chengdu Campus, Chengdu 611130, China

**Keywords:** rabbit, colon, diarrhea, microorganism, transcriptome, untargeted metabolomics

## Abstract

**Simple Summary:**

After being fed an antibiotic-free diet, some rabbits showed typical diarrhea symptoms. In order to explore the reasons, this study used multiple omics analysis. *Bacteroidetes* and *Proteobacteria* were significantly upregulated in the colon of diarrhea rabbits, and the ratio of *Firmicutes* to *Bacteroidetes* was decreased. The significantly upregulated differential genes were mainly enriched in the IL-17 signaling pathway and were involved in promoting inflammatory response. The different metabolites were mainly enriched in tryptophan metabolism and bile secretion, which affected the anti-inflammatory function. In addition, *Bacteroides* is positively correlated with 4-Morpholinobenzoic acid and Diacetoxyscirpenol, which is believed to be an important cause of inflammation. The enrichment of *Proteobacteria* is also related to the high expression of the IL-17 signaling pathway.

**Abstract:**

Diarrhea symptoms appeared after antibiotics were banned from animal feed based on the law of the Chinese government in 2020. The colon and its contents were collected and analyzed from diarrheal and healthy rabbits using three omics analyses. The result of the microbial genomic analysis showed that the abundance of *Bacteroidetes* and *Proteobacteria* increased significantly (*p*-value < 0.01). Transcriptomes analysis showed that differentially expressed genes (DEGs) are abundant in the IL-17 signaling pathway and are highly expressed in the pro-inflammatory pathway. The metabolome analysis investigated differential metabolites (DMs) that were mainly enriched in tryptophan metabolism and bile secretion, which were closely related to the absorption and immune function of the colon. The results of correlation analysis showed that *Bacteroidetes* was positively correlated with 4-Morpholinobenzoic acid, and 4-Morpholinobenzoic acid could aggravate inflammation through its influence on the bile secretion pathway. The enriched DMs L-Tryptophan in the tryptophan metabolism pathway will lead to the functional disorder of inhibiting inflammation by affecting the protein digestion and absorption pathway. Thus, the colonic epithelial cells were damaged, affecting the function of the colon and leading to diarrhea in rabbits. Therefore, the study provided an idea for feed development and a theoretical basis for maintaining intestinal tract fitness in rabbits.

## 1. Introduction

Antibiotics have played an important role in animal feed over the past half-century. It was used to prevent and treat animal infection for a long time, expedite intestinal absorption and digestion, and expedite its production capacity [1]. Long-term addition of antibiotics in feed, but the problem also followed [2]. Long-term feeding the feed containing antibiotics usually leads to drug resistance in animals [3,4]. Moreover, more and more studies proved that the residue and accumulation of antibiotics in all kinds of meats would affect the health of consumers [5]. An increasing number of people are paying attention to the rational use of antibiotics. China’s Ministry of Agriculture and Rural Affairs signed a national law banning antibiotics in 2020. Then, typical symptoms of diarrhea appeared in some individuals, and the mortality and feeding cost of rabbitry increased [6].

The intestinal tract is an important organ for defense and immune response, as well as the key to nutrient absorption [7,8]. The colon has a special structure and the highest pH, which also leads to a more complex internal environment of the colon tissue [9]. Inflammation of the colon can cause diarrhea [10] and could damage the mucosal structure of the colon epithelium [11]. Damage to the colon structure could affect the composition of the microbial community and then affect the normal absorption function of the colon, resulting in metabolic disorders [12,13].

Multi-omics research methods can reveal the complex process of rabbit nutrient absorption in a more three-dimensional way [14,15]. Thus, this study aimed to explore the mechanism of diarrhea in rabbits caused by colon lesions. The colon tissues and contents were collected and analyzed by microbial group, transcriptome, and non-targeted metabolism group. It provided an idea for the development of healthy feed and a theoretical basis for further study of rabbit diarrhea caused by colonic inflammation.

## 2. Materials and Methods

### 2.1. Ethics Statement

The experimental procedures in this study have been approved by the Animal Care and Utilization Committee from the College of Animal Science and Technology, Sichuan Agricultural University, China.

### 2.2. Animals Feeding Condition

The project was performed in the rabbitry, Leibo County (103.57° E, 28.26° N), Sichuan Province, China. The rabbits were raised in clean cages with regularly inoculating vaccines. The feed was purchased from a local feed manufacturer. July 2021, the rabbit farm began to use the antibiotic-free diet based on the requirements of national policy. The rabbits appeared to have typical diarrhea symptoms, such as loss of appetite, feces not forming, and feces stench.

### 2.3. Samples Collection

40-day-old weaned rabbits were selected. Six female rabbits with typical diarrhea symptoms were selected as the diarrhea group (DIA), and six female rabbits without diarrhea symptoms were selected as the control group (CON). The selected rabbits were fasted for 24 h and were slaughtered by the electric bloodletting method. The colon tissue and its content were taken immediately and were preserved in liquid nitrogen at −80 °C. The RNA and DNA in samples were extracted by conventional methods and sent to Novogene Bioinformatics Technology Co., Ltd. (Beijing, China) for sequencing and preliminary analyses [15,16].

### 2.4. Morphological Section Analysis of Rabbit Colon

Some colon tissues were washed with normal saline, fixed with neutral formalin, dehydrated with ethanol, embedded in paraffin, sectioned, and stained with hematoxylin and eosin (HE). The histopathological features, using a CX22 microscope (OLMPUS, Tokyo, Japan), were observed and photographed using Leica microscopic imaging system (DM1000, Leica, Wetzlar, Germany).

### 2.5. Microbial Genomic16S rRNA Gene Sequencing and Sequencing Analysis

After DNA amplification, purification, and construction of the SMART bell library, the bam file was exported by the SMRT analysis software of PacBio [17]. OTUs (operational taxonomic units) clustering and species classification analysis were performed after distinguishing samples according to barcode [18,19]. The representative sequences of each OTU were annotated to obtain the corresponding species information and relative abundance of species. The diversity and richness of microbial communities in the samples were analyzed by alpha diversity analysis. Qiime software (Version1.9.1, Novogene, Beijing, China) was used to calculate the alpha diversity index (Shannon index, chao1 index, and Simpson index), R software was used to analyze the difference between groups of alpha diversity index, and the Wilcox rank-sum test was used for comparison. The microbial communities between the two groups of samples were compared and analyzed by beta diversity analysis. The species annotation results of the two groups and the abundance information of OTUs were combined, and then the weighted unifrac distance was calculated and constructed through the abundance relationship between OTUs of the same classification in different groups.

Finally, multivariate statistical methods are used to enter non-metric multi-dimensional scaling (NMDS) and multi-response permutation procedure (MRPP). The microorganisms with significant differences between groups at each classification level were found by *t*-test (*p*-value < 0.05). LEfSe was used to find high-dimensional biomarkers, and the microbial population with the most significant difference was determined with “LDA score >4” as the standard.

### 2.6. RNA-Seq Data and Differential Expression Analysis

After the extracted RNA was screened, amplified, and purified, the library was finally obtained using NEBNext^®^ UltraTM Directional RNA library Prep Kit for Illumina^®^ (San Diego, CA, USA) [20]. The sequencing fragment image data measured by a high-throughput sequencing instrument was transformed into sequence data (reads) by CASAVA base recognition to obtain clean reads. Q20, Q30, and CG of clean readings are also calculated. In the correlation analysis of data between samples, the R2 of the Pearson correlation coefficient is greater than 0.92 (ideal sampling and experimental conditions). In the specific operation of the project, R2 between biological replicate samples is at least greater than 0.8. Otherwise, samples need to be properly explained or retested. According to the FPKM values of all genes in each sample, the correlation coefficients of intra-group and inter-group samples are calculated, and the heat map is drawn, which can visually display the differences between groups and the repetition of samples within groups. The higher the correlation coefficient between samples, the closer the expression pattern.

Then DESeq2 software (1.20.0) was used to analyze the differential expression of the two comparative combinations. The *p*-value ≤ 0.05, and the FDR ≤ 0.01. It was considered that the genes with |log2 (fold change)| >1 were significantly differentially expressed genes, and the results were shown in the voLithocholic acidnic map. Subsequently, GO classification function annotation and KEGG enrichment analysis were performed on these DEGs to obtain further differential enrichment results.

### 2.7. Metabolome Analysis

The metabolites in the intestinal segment were studied based on LC-MS technology. After liquid nitrogen grinding and centrifugation of intestinal tissue samples, the supernatant was injected into the ultra-performance liquid chromatography-tandem mass spectrometry (UHPLC-MS/MS) system for analysis [21]. Firstly, the raw data of mass spectrometry were imported into Compound discoverer 3.1 software for spectral processing and database retrieval, and the qualitative and quantitative results of metabolites were obtained. Then, the quality of data was controlled to ensure the accuracy and reliability of the data. Using high-resolution mass spectrometry (HRMS) technology, we can make the non-target metabolic group as much as possible to detect the molecular characteristic peaks in the sample. The raw data after offline is preprocessed by CD3.1 data processing software. In order to make the identification accurate, we extract the peaks according to the set of ppm, signal-to-noise ratio (S/N), additive ions, and other information and quantify the peak area. Then mzCloud, mzVault, and MassList databases were compared to identify metabolites. Finally, metabolites with a coefficient of variation of less than 30% in QC samples were retained as the final result. The metabolites were compared with KEGG, HMDB, and other databases to obtain the annotation results. Then, a multivariate statistical analysis of metabolites was performed, including principal component analysis (PCA), partial least squares discriminant analysis (PLS-DA), and other methods to establish the relationship between the expression of metabolites and samples [22]. According to the results of Q2 and R2, the model was judged to reveal the differences in metabolic patterns between different groups. KEGG enrichment pathway analysis was performed on the differential metabolites to obtain clearer and more detailed differential analysis results.

### 2.8. Association Analysis

The association analysis was performed on the composition of colony differences and the results of differential metabolites obtained from 16S rDNA and metagenomic analysis of genus-level differences to compare the association degree between species diversity and metabolites in environmental samples. The value between (−1,1) is the correlation coefficient, which is negatively correlated when the correlation coefficient is less than 0, and positively correlated when the correlation coefficient is greater than 0. The Pearson correlation coefficient rho and *p*-value of top10 and top20 were calculated, and the scatter plot analysis was drawn with the results of |rho| ≥0.6 and *p*-value ≤ 0.05, which intuitively showed the results of the correlation scatter plot analysis of the expression of different bacteria and different metabolites.

The top100 of genes with significant differences obtained by transcriptome analysis and the top50 of metabolites with significant differences obtained by metabolic progenitor analysis were analyzed based on the Pearson correlation coefficient to compare the correlation between the two. The results are presented in the clustering heat map. When the correlation coefficient (−1,1) is greater than 0, it is positive and negative when the correlation coefficient is less than 0. Then all the differential genes and differential metabolites were compared with the KEGG pathway database at the same time to obtain their common pathway results and determine the main biochemical pathways and signal transduction pathways that differential metabolites and differential genes participate in together.

## 3. Results

### 3.1. Colon Tissue Sections

Colon tissue samples stained with HE are shown in Figure 1. In the DIA group, the colonic mucosa epithelium was exfoliated, part of the intestinal wall was necrotic, and the number of lymphocytes was decreased, presenting typical that is diarrhea. In contrast, the rabbit of the CON group had intact colon structure and no pathological features.

### 3.2. Microbial Community Imbalance

After alpha diversity analysis, the results are shown in Table 1. All the alpha diversity indices in the DIA group were reduced, and the Shannon index and Chao1 index were significantly reduced (*p*-value < 0.05), indicating that the richness and evenness of microbial communities in the DIA group were significantly reduced (*p*-value < 0.05). The results of MRPP analysis also showed that the microbial separation between the two groups was obvious.

The results of PCoA and NMDS were consistent with the above results. The expected-delta value and A value jointly indicated that the difference between groups was greater than that within the group, and the significance < 0.05 indicated that the difference was significant (Table 2).

The relative abundance differences of microorganisms between the two groups were compared (Figure 2), and a *t*-test was performed to identify the species with significant differences (*p*-value < 0.05) between groups (Table 3). The results showed that *Bacteroidetes* were the dominant bacteria in the DIA group at the levels of phylum, class, order, family, and genus, with an average proportion of 42%. In addition, *Proteobacteria* was also significantly enriched in the DIA group, and the difference was significant (*p*-value < 0.01). Some endemic microorganisms, *Peptostreptococcaceae*, Synergistetes, and Cyanobacteria, were detected in the DIA group. *Firmicutes*, *Clostridia*, and *Ruminococcaceae* were the dominant flora in the CON group, and the differences were significant (*p*-value < 0.01). It was also found that *Bacteroidetes* and *Firmicutes* were the main flora at the phylum level, but *Bacteroidetes* > *Firmicutes* in the DIA group and the opposite in the CON group. Moreover, the dominant flora at other levels fall under the classification of these two microorganisms.

LEfSe was used to analyze the differences in microbial communities between the two groups at different classification levels. The results are shown in Figure 3. In the DIA group, *Bacteroidetes* were significantly enriched at all levels, and *Proteobacteria* were also significantly enriched at the phylum level. The analysis results of the CON group showed that *Ruminococcaceae*, *Clostridia*, *Clostridiales* and *Firmicutes* were significantly enriched.

### 3.3. Differential Expression of Genes in Colon

After raw data filtering, sequencing error rate inspection, and GC content distribution test. In the DIA group, 44,623,012 raw reads and 43,329,060 clean reads were read on average. The average Q20 of clean reads was 97.67%, and the average Q30 was 93.65%. Then, 45,259,949 raw reads and 44,084,909 clean reads were read in the CON group, with an average Q20 of 97.67% and Q30 of 93.67%. The results of correlation analysis between samples (Figure 4) showed that when the correlation coefficient R2 > 0.8, it was considered that the correlation between samples was high. The general correlation of the samples in the CON group was high, with an average R2 of 0.892, while that in the DIA group was 0.706. The sample correlation between the DIA group and the CON group is low, with an average R2 of 0.773.

A total of 21,703 DEGs were obtained, as shown in the volcano map (Figure 5). There were 321 DEGs with significant indigenous differences, 131 DEGs were upregulated, and 190 DEGs were downregulated. The significantly upregulated DEGs included S100A9, S100A8, MMP1, CXCL8, and other DEGs, while significantly downregulated DEGs included ITGA11, FN1, COL6A6, COL1A1, and other DEGs. The result of GO enrichment analysis indicated that DEGs were mainly enriched in extracellular space, extracellular matrix, and collagen-containing extracellular matrix. The results after GO enrichment analysis and KEGG enrichment analysis of these DEGs were shown (Figure 6 and Figure 7). KEGG enrichment analysis indicated that significantly upregulated DEGs were mainly enriched in the IL-17 signaling pathway. The downregulated DEGs were mainly concentrated in protein digestion and absorption, ECM-receptor interaction, and focal adhesion.

### 3.4. Differential Metabolite Analysis

The results of metabolite quantitative analysis showed that a total of 1663 metabolites were collected, including 1191 positive metabolites and 472 negative metabolites. The results of the QC sample correlation score (Figure 8) showed that the correlation R2 of both negative and positive metabolites was between 0.989 and 0.992, indicating that the correlation between samples was high and the data accuracy was high. PCA analysis was performed on all samples, and the results of PCA analysis are shown in Figure 8. The cathodic and anodic metabolites of the two groups were significantly separated, and the dispersion of the DIA group was higher. This indicated that diarrhea leads to more complex metabolites in the DIA group. The results of the PLS-DA analysis were consistent with the above results. After screening for differential metabolites, the results showed in a volcano map (Figure 9). There were 651 differential metabolites, including 472 for positive and 179 for negative, and 373 metabolites were significantly upregulated, 278 metabolites were significantly downregulated, 194 metabolites were significantly downregulated, and 84 metabolites were significantly downregulated.

The results were presented as KEGG pathways, and the main biochemical metabolic pathways and signal transduction pathways involved in differential metabolites could be determined. Results of the KEGG pathway were shown (Figure 10). The results showed that the differential metabolites at the positive were mainly enriched in tryptophan metabolism (*p*-value = 0.013), phenylalanine, tyrosine and tryptophan biosynthesis (*p*-value = 0.06), cortisol synthesis and secretion (*p*-value = 0.06), Cushing’s syndrome (*p*-value = 0.06). The differential metabolites of the negative were mainly enriched in bile secretion (*p*-value = 0.20), biosynthesis of acid (*p*-value = 0.24) and biosynthesis of unsaturated fatty acids (*p*-value = 0.26).

### 3.5. Metabolite-Centric Correlation Analysis

Figure 11 shows the results of correlation analysis between DMs and microorganisms. Under genus-level classification, dominant *Bacteroides* in DIA group and Diacetoxyscirpenol (rHO = 0.80, *p*-value < 0.01) and 4-morpholinobenzoic acid (rHO = 0.71, *p*-value < 0.01) in positive, 2-(1-{2- [(3-Furylmethyl) Amino]-2-oxoethyl}cyclohexyl)acetic acid (RHO = 0.89, *p*-value < 0.01) and 15(R) -lipoxin and A4 (RHO = 0.77, *p*-value < 0.01) in negative has a strong positive correlation and is extremely significant. The dominant bacteria in CON group *Ruminococcaceae* were compared with Robenidine (RHO = 0.92, *p*-value < 0.01) in positive and INH (RHO = 0.89, *p*-value < 0.01) in negative.

KEGG pathway comparative analysis was performed on DEGs and DMs. The results of the KEGG pathway analysis (Figure 12) showed that the enrichment in the Arachidonic acid metabolism pathway was extremely significant (*p*-value < 0.01). Enrichment in protein digestion and absorption pathways was highly significant (*p*-value < 0.01).

## 4. Discussion

After being transferred to an antibiotic-free diet, we found that some rabbits developed typical diarrhea symptoms such as loss of appetite, dilute feces, and feces stench. The HE-stained intestinal tissue samples showed that damage in the colon was caused by an inflammatory response. Inflammation would lead to abnormal immune response and damage to intestinal epithelial mucosa, thus affecting their functions [11,23].

The diversity and evenness of microorganisms in the DIA group decreased significantly, while *Bacteroidetes* increased significantly and *Firmicutes* decreased significantly. The ratio of *Firmicutes* and *Bacteroidetes* could often reflect the health situation in the colon [24,25]. In healthy individuals, the index of *Firmicutes*/*Bacteroidetes* (F/B > 1) is usually high, and an imbalance in the F/B index can lead to an inflammatory response [26,27]. The same phenomenon was found in this study. Individuals with a more balanced index of *Firmicutes*/*Bacteroidetes* often have a better ability to absorb nutrients [28,29]. *Bacteroidetes* plays a role in assisting the absorption of sugars and lipids, which is a kind of Probiotics. However, *B**acteroidetes* accumulate in large quantities and become pathogenic bacteria [30,31]. In addition, *P**roteobacteria* were significantly enriched in DIA. Studies have shown that there is no significant enrichment in healthy individuals [32]. *P**roteobacteria* enrichment causes an inflammatory response that leads to the abnormal function of colon epithelial cells [33,34,35]. Thus, speculated that *Bacteroidetes* and *Proteobacteria* may be an important cause of colon inflammation. Because there is a reciprocal balance between microbes and hosts in healthy individuals, changing the feed formula to adjust the balance of microbial communities mainly composed of *Firmicutes* and *Bacteroidetes* can be a way to prevent diarrhea and maintain animal health [26,36].

We found that a large number of significantly upregulated DEGs were enriched in the IL-17 signaling pathway. The interleukin-17 (IL-17) cytokine family, mostly produced by Th17 cells, plays a major role in colon inflammation in mice and humans [37,38]. IL-17a expression has been implicated in many immune diseases and inflammatory responses [39]. IL-17A could activate on a variety of cell types, and the high expression of IL-17A could directly promote the production of pro-inflammatory molecules, so the expression of IL-17A is strictly regulated by the IL-17 signaling pathway [40,41]. S100A9, MMP1, CXCL8, and S100A8 DEGs in the IL-17 signaling pathway were also found and significantly upregulated. Meanwhile, significantly downregulated DEGs were concentrated in protein digestion and absorption, ECM-receptor interaction, and focal adhesion. Moreover, the three pathways were very important and responsible for maintaining intestinal epithelial cell structure and immune response, including some anti-inflammatory functions. Studies showed that inflammation would break the homeostasis of ECM and weaken its ability to be anti-inflammatory, which will further lead to lesions [42,43]. The main function of focal adhesion is to induce neutrophils to relieve inflammation, and neutrophils play a major role in regulating inflammation and tissue repair. Low expression of focal adhesion pathways could lead to neutrophil dysfunction and greatly reduce the inhibitory effect of inflammation [44,45].

After differential enrichment analysis of DMs showed that tryptophan metabolism and bile secretion was the most significant. The metabolites in tryptophan metabolism included indole-3-acetic acid, L-tryptophan, Indole, L-kynurenine, etc. Trp is a biologically essential amino acid that plays an important role in the growth, development, and reproduction of mammals. Moreover, it was an important precursor for the synthesis of metabolites used for neurotransmitters involved in immune and inflammatory responses [46,47]. Studies showed that indole-3-acetic acid, L-tryptophan, and indole could effectively improve the damage of inflammation to the colon. Moreover, l-tryptophan could also play a role in the NF-κB signaling pathway, which could block transcription and activation of pro-inflammatory cytokines, and other studies showed that it can reduce the risk of colorectal cancer [48]. Metabolites enriched in the bile pathway were associated with regulating carbohydrate and lipid absorption and energy metabolism [49]. Bile acids (BA) could promote the absorption of vitamins in the intestine and improve the immune function of the body to maintain individual health [50,51]. Lithocholic acid and deoxycholic acid could regulate bile levels and hepatoenteric circulation by activating the Farnesoid X receptor (FXR). One of the characteristics of colonic inflammation is also the decreased expression level of FXR, which increases the pro-inflammatory appearance and the generation of oxidative stress by changing the BA receptor level. It even affects liver cells [52,53]. Meanwhile, lithocholic acid has the function of protecting the epithelial barrier, which plays an important role in inhibiting inflammation [53,54]. It was found that BA-related metabolites were significantly downregulated, and the end of the correlation analysis showed that *Bacteroidetes* were negatively correlated with these metabolites. Therefore, it was concluded that the enrichment of *Bacteroidetes* reduced the expression level of the bile pathway, which led to the decrease in digestion and absorption level of colon, weakened its inflammatory inhibition function, and further aggravated the inflammatory reaction [54,55].

Individual metabolic health is closely related to microbial levels. In this study, 4-morpholinobenzoic acid, Diacetoxyscirpenol, and *Bacteroides* are positively correlated (*p*-value < 0.01) and were enriched in related pathways. In conclusion, we speculated that the enrichment of *Bacteroidetes* was the main cause of colon inflammation. On the one hand, protein digestion and absorption, ECM-receptor interaction, and focal adhesion were affected, resulting in downregulated expression of these pathways, which weakened nutrient absorption, immune function, and anti-inflammatory effect. These reasons lead to structural damage to the colon epithelium. On the other hand, the IL-17 signaling pathway and other pro-inflammatory pathways were activated, further exacerbating the inflammatory response.

## 5. Conclusions

The result showed that the abundance of *Bacteroidetes* and *Proteobacteria* significantly increased (*p*-value < 0.05), while *Firmicutes* significantly decreased (*p*-value < 0.05). The pathway of bile secretion and tryptophan metabolism were enriched with the most DMs, which impaired the absorption and energy conversion functions of the colon, and affected its immune and anti-inflammatory functions. The high expression of the IL-17 signaling pathway aggravates the inflammatory response. We speculated that the main cause might be the chain reaction caused by the imbalance of *Bacteroidetes* and *Firmicutes*. This leads to an inflammatory response and damage to the epithelial structure of the colon. Moreover, nutrient absorption, anti-inflammatory, and other functions were impaired. Under the common action of these factors, the rabbits showed typical diarrhea symptoms. Studies showed that proper adjustment of feed formula to restore microbial ecological balance in the colon may effectively prevent diarrhea. However, its specific mechanism needs further research and verification. The study provided an idea for feed development and a theoretical basis for maintaining intestinal tract fitness in rabbits.

## Figures and Tables

**Figure 1 animals-12-01497-f001:**
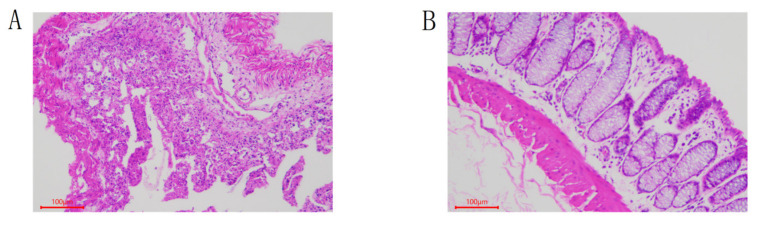
Pathological observation of colon tissue in rabbits by microscope (HE-staining, 100×). (**A**) DIA. (**B**) CON.

**Figure 2 animals-12-01497-f002:**
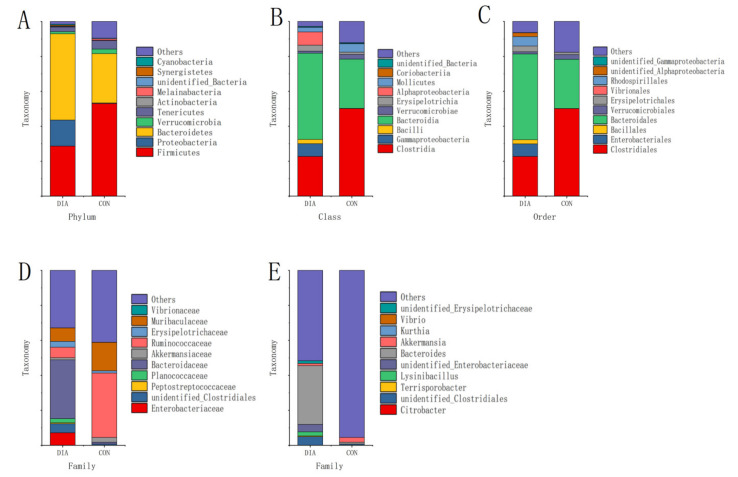
Average relative abundances histogram of species at phylum level (**A**), class level (**B**), order level (**C**), family level (**D**), genus level (**E**).

**Figure 3 animals-12-01497-f003:**
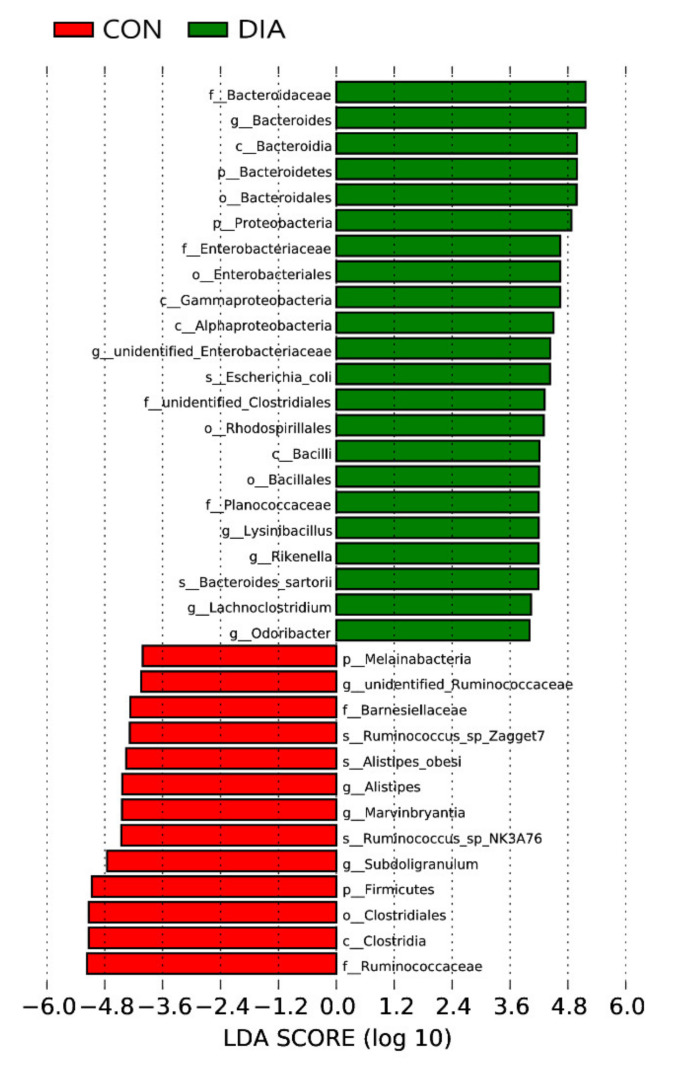
LDA score of LEfSe-PICRUSt.

**Figure 4 animals-12-01497-f004:**
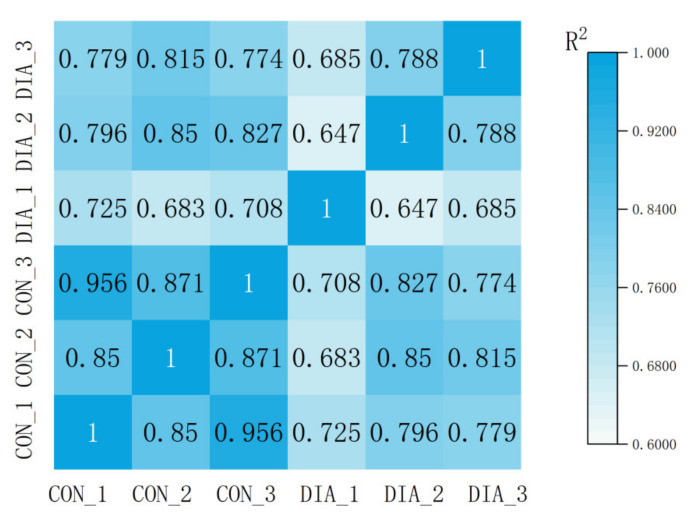
Heat map of correlation analysis between samples. The higher the correlation coefficient between samples, the closer the expression pattern is.

**Figure 5 animals-12-01497-f005:**
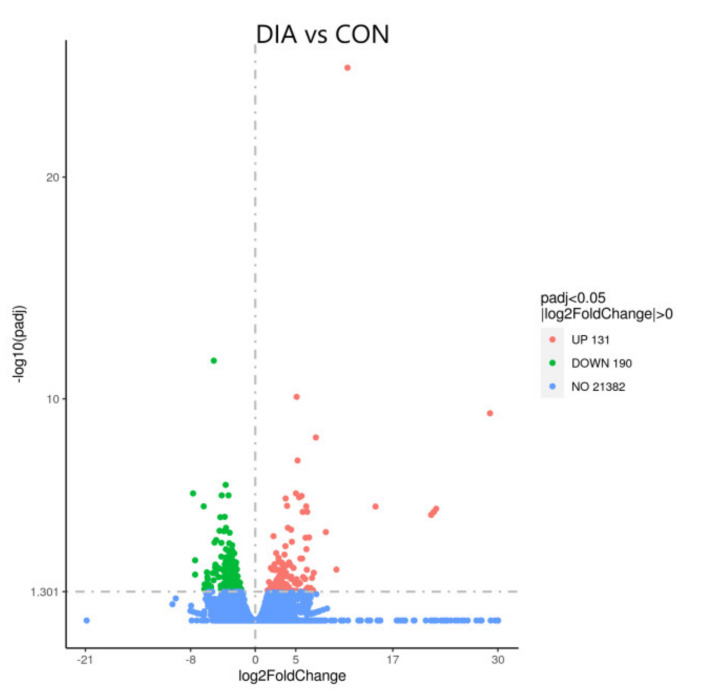
Volcano-map of DEGs in colon tissue of DIA and CON.

**Figure 6 animals-12-01497-f006:**
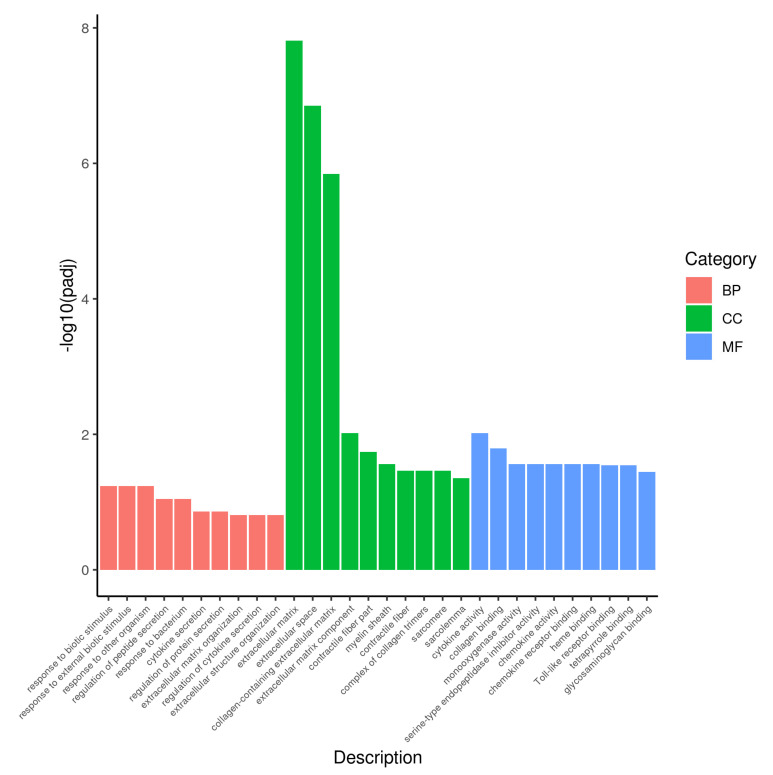
The most enriched GO terms and pathway of the DEGs.

**Figure 7 animals-12-01497-f007:**
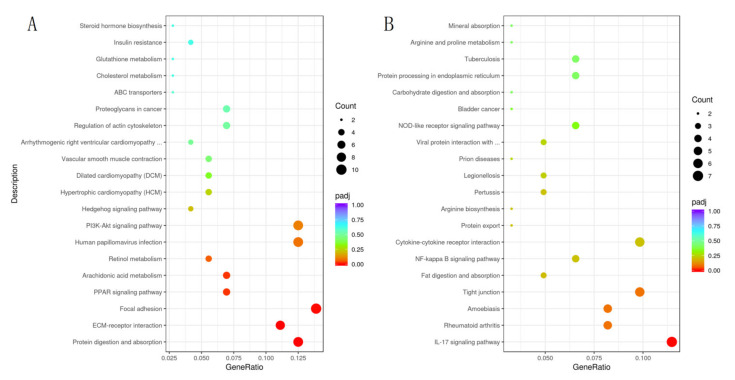
The most enriched KEGG pathway of the DEGs, significantly upregulated DEGs (**A**), significantly downregulated DEGs (**B**).

**Figure 8 animals-12-01497-f008:**
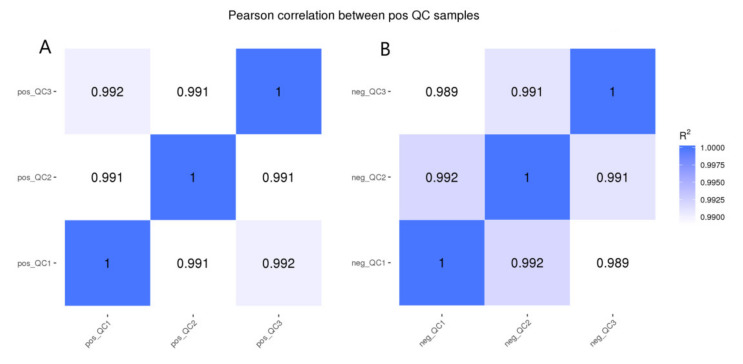
QC sample correlation score. (**A**) is the QC sample correlation analysis results of positive metabolites, (**B**) is the QC sample correlation analysis results of negative metabolites.

**Figure 9 animals-12-01497-f009:**
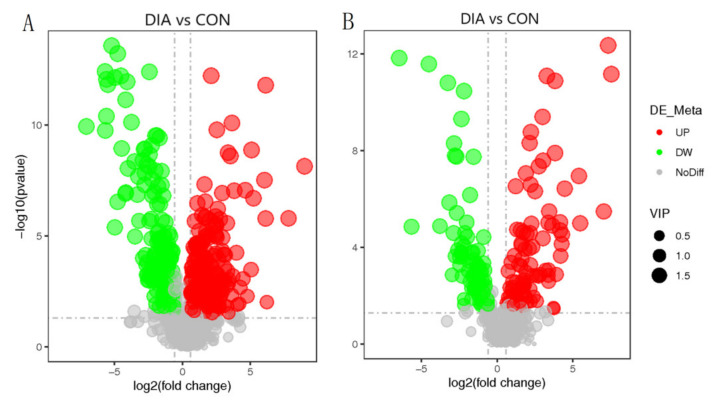
Volcano map of DMs in colon tissue of DIA and CON. Positive DMs (**A**), negative DMs (**B**).

**Figure 10 animals-12-01497-f010:**
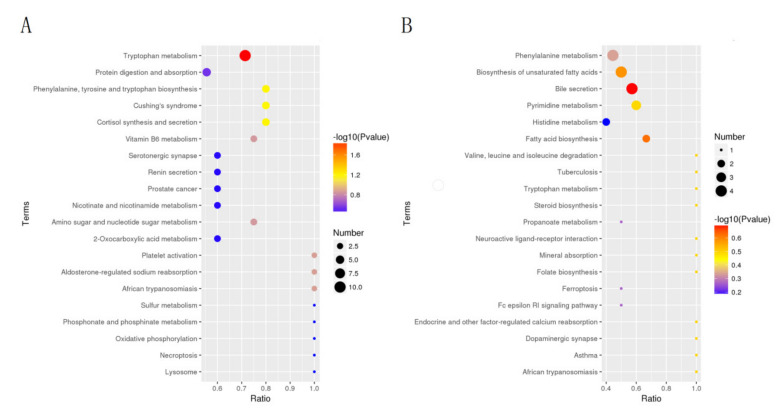
The most enriched KEGG pathway of the DMs. Positive DMs (**A**), negative DMs (**B**).

**Figure 11 animals-12-01497-f011:**
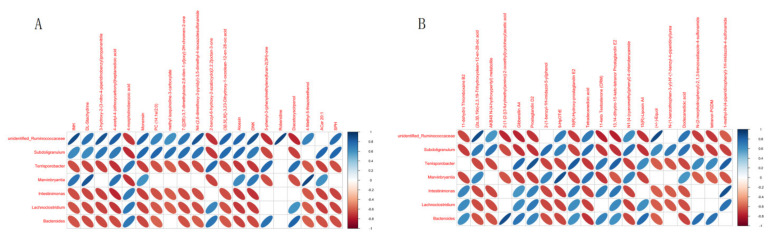
Correlation analysis results of differential-metabolites top20 and 16S microbial-genomic Top10. Positive DMs (**A**), negative DMs (**B**).

**Figure 12 animals-12-01497-f012:**
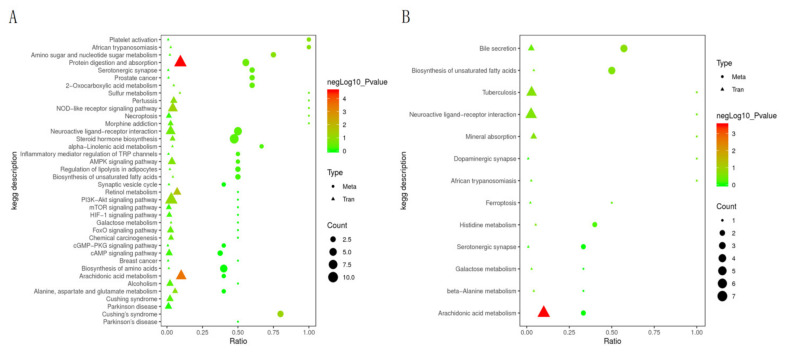
KEGG enrichment analysis of DMs and DEGs. Positive DMs (**A**), negative DMs (**B**).

**Table 1 animals-12-01497-t001:** Alpha diversity analysis of intestinal microbial of rabbit colon. The notation * indicate significant differences (*p* < 0.05) among groups.

Group	Species	Shannon	Simpson	Chao1	Coverage
DIA	89	4.06	0.812	124.49	0.96
CON	146	5.44 *	0.936	215.25 *	0.93

**Table 2 animals-12-01497-t002:** MRPP analysis of intestinal microbial of rabbit colon.

Group	A	Observed-Delta	Expected-Delta	Significance
CON.VS.DIA	0.2053	0.6475	0.8147	0.002

**Table 3 animals-12-01497-t003:** *T*-test analysis of rabbit colon intestinal microflora in DIA group and CON group at phylum, class, order, family, and genus level.

Level	Name	avg(DIA)	sd(DIA)	avg(CON)	sd(CON)	*p*-Value
phylum	*Firmicutes*	0.286470	0.083302	0.530045	0.110976	0.001857
	*Proteobacteria*	0.149093	0.078586	0.003590	0.003954	0.006152
	*Bacteroidetes*	0.493764	0.147044	0.281935	0.083914	0.015600
	*Melainabacteria*	0.000377	0.000585	0.007936	0.006042	0.027750
class	*Clostridia*	0.227135	0.067548	0.500944	0.127411	0.001879
	*Bacteroidia*	0.493764	0.147044	0.281935	0.083914	0.015600
order	*Bacteroidales*	0.025132	0.038578	0.281746	0.083690	0.000243
	*Verrucomicrobiales*	0	0	0.027021	0.016279	0.009673
family	*Peptostreptococcaceae*	0.005480	0.003822	0	0	0.017071
	*Bacteroidaceae*	0.335978	0.262398	0.010959	0.009849	0.028913
	*Ruminococcaceae*	0.059901	0.039701	0.368102	0.103267	0.000356
genus	*Terrisporobacter*	0.004535	0.003286	0	0	0.019661
	*Bacteroides*	0.335978	0.262398	0.010959	0.009849	0.028913
	*Lachnoclostridium*	0.021542	0.016209	0	0	0.022563
	*unidentified_Ruminococcaceae*	0.002834	0.003344	0.012093	0.005203	0.005681
	*Intestinimonas*	0.003590	0.002626	0	0	0.020366
	*Subdoligranulum*	0	0	0.001322	0.000853	0.012676
	*Marvinbryantia*	0	0	0.001889	0.001373	0.019868

## Data Availability

All the figures and tables used to support the results of this study are included.

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
