# Peer review of "The Multi-Omics Analysis Revealed Microbiological Regulation of Rabbit Colon with Diarrhea Fed an Antibiotic-Free Diet"

_animals, 2022, doi:10.3390/ani12121497_

Round 1
Reviewer 1 Report
Accept in present form.
Reviewer 2 Report
The authors have revised the manuscript very carefully and taken the comments into account accordingly. It can be accepted in its present form.
This manuscript is a resubmission of an earlier submission. The following is a list of the peer review reports and author responses from that submission.
Round 1
Reviewer 1 Report
The article deals with multi-omics analysis in rabbit with diarrhea, focusing on microbiological regulation.
It is interesting paper in which the authors are dealing with the microbial genomic analysis, transcriptomes analysis, the metabolome analysis and also, they made the pathological observation of caecum of rabbit with diarrhea. Compared to control group, in each group was six 40 days old weaned female rabbits, the colonic mucosa epithelium was exfoliated and part of intestinal wall was necrotic. It is obvious that that those rabbits had diarrhea. Rabbits received feed without antibiotics, since China government banned the use of antibiotics from 2020 and it is a relatively new issue in China.
This study or results of this study are another part of the study: “The Multi-omics analysis Revealed a Metabolic Regulatory 2 System of Cecum in Rabbit with Diarrhea in which have the same rabbits, but different approach.”
The results of this study is that some of the microorganisms are more presented in the colon of rabbits with diarrhe and others in the control group, so the first one were up-regulated and the second one were down-regulated in diarrhea group. They also found some differences in IL-17 signalling pathway, up-regulated in diarrhea group, which is involved in promoting inflammation.
The paper is well written, understandable with admissible English language.
I do not have any suggestions for changes in article, except some small mistakes:
- In page 2, second paragraph it is pH written wrong an after that the comma must be without the interval
- Page 12, first paragraph, dot after the second sentence must be without the interval.
- Page 12, third paragraph, IL-17 and after that it is in the parenthesis written the long version, this must be done, first time that IL-17 is mentioned in the text.
- Page 12 IL-12a and IL-12A, the authors must choose, one of the versions.
- Page 13, second paragraph. There is something missing at the end of first sentence: ….and Bile secretion was.
- Page 13 second paragraph, what is the meaning of BA?
Author Response
Thank you very much for the reviewer’s comments. We have carefully corrected all the mistakes pointed out by reviewer A.
Reviewer 2 Report
This study deals with a highly topical issue. On the one hand, because only a few papers are available on the rabbit microbiome, and on the other hand, because the topic of dispensing with antibiotics is addressed.
The present data were obtained in clinically healthy animals and animals with diarrhoea. Unfortunately, the authors fail to define exactly what diarrhoea is for them. Was the dry matter content in the faeces determined? Could pathogenic germs be found in the faeces? Coccidia? Especially the last point is of crucial importance for the results and their interpretation. Pathogenic germs inevitably lead to a shift in the microbiota.
Have the animals been clinically examined? An increase in temperature, which is indicative of fever, would support the infection. It would also have consequences for the immune system and its parameters.
In addition to these problems of content, it must be pointed out that the manuscript contains a great many orthographical errors and that the English is in urgent need of revision.
Notes
See attachment

Author Response
Comment 1: The present data were obtained in clinically healthy animals and animals with diarrhoea. Unfortunately, the authors fail to define exactly what diarrhoea is for them. Was the dry matter content in the faeces determined? Could pathogenic germs be found in the faeces? Coccidia? Especially the last point is of crucial importance for the results and their interpretation. Pathogenic germs inevitably lead to a shift in the microbiota.
Response: Thanks very much for the reviewer’s suggestion. These animals were not clinical animals, they were from rabbit farm. This was the problem we found in production, that was, after feeding an antibiotic-free diet, rabbits had diarrhea. We found that some rabbits appeared such as loss of appetite, or dilute feces, faeces stench. These are typical manifestations of diarrhea. After the rabbits were slaughtered, we collected rabbit colon contents for metabolites and microbial analysis, rather than feces. This was more conducive to focusing on colon lesions. After analysis, Bacteroidetes and Proteobacteria were highly enriched in DIA group. We think these were the main pathogenic germs.
Comment 2: Have the animals been clinically examined? An increase in temperature, which is indicative of fever, would support the infection. It would also have consequences for the immune system and its parameters.
Response: Thanks very much for the reviewer’s suggestion. These animals did not undergo clinical examination. These were just commodity rabbits in our rabbit farm, not clinical animals. This study was found this phenomenon in production for research, not designed experiments. Temperature factor we considered, but later did not retain this reference factor. Because there were
many changes in body temperature, it is difficult to determine that diarrhea is directly caused. And found diarrhea and collect samples was in summer, can not exclude climatic factors. We could ensure the accuracy of this study by using microorganism、transcriptome、metabolomics. in the colon as the main reference.
Comment 3: Some questions about the attachment.
Response: Thanks very much for the reviewer’s suggestion and comment. The orthographical errors in the manuscript have been corrected and marked with yellow highlight.
(Line 62 and 64 in pages 2) Thus, this study aimed to explore the mechanism of diarrhea in rabbits caused by colon lesions. The colon tissues and contents were collected and analyzed by microbial group, transcriptome and non-targeted metabolism group. It provided an idea for the development 2 of healthy feed and a theoretical basis for further study of rabbit diarrhea caused by colonic
inflammation.
(Line 92 in pages 2) The histopathological features, using CX22 microscope (OLMPUS, Japan), were observed and photographed using Leica microscopic imaging system (DM1000, Leica, Germany).
(Line 143 in pages 3) Then, quality of data were controlled to ensure the accuracy and reliability of the data.
(Line 317 in pages 14) After being transferred to an antibiotic-free diet, we found that some rabbits developed typical diarrhea symptoms such as loss of appetite, or dilute feces, faeces stench.
(Line 391 in pages 15) These reasons lead to Structural damage to the colon epithelium.
Thanks again to the Reviewer #2, your suggestion and comment are very helpful to me.